# Maintenance of service delivery during medical countermeasures deployment: The association between the COVID-19 vaccine rollout and continuity of routine childhood immunization services in Uganda

Steven Ndugwa Kabwama[1,2]*, Neda Razaz[3], John M. Ssenkusu[4], Helena Lindgren[5,6], Rhoda K. Wanyenze[7], Alfred Driwale[8], Tobias Alfvén[1,9]

1 Department of Global Public Health, Karolinska Institutet, Stockholm, Sweden, 2 Department of Community Health and Behavioral Sciences, Makerere University School of Public Health, Kampala Uganda, 3 Clinical Epidemiology Division, Department of Medicine, Solna, Karolinska Institutet, Stockholm, Sweden, 4 Department of Epidemiology and Biostatistics, Makerere University School of Public Health, Kampala, Uganda, 5 Department of Women's and Children's Health, Karolinska Institutet, Stockholm, Sweden, 6 Department of Health Promotion, Sophiahemmet University, Stockholm, Sweden, 7 Department of Disease Control and Environmental Health, Makerere University School of Public Health, Kampala Uganda, 8 Department of Institutional Capacity Building and Human Resources for Health Development, Ministry of Health, Kampala, Uganda, 9 Sachs' Children and Youth Hospital, Stockholm, Sweden

* steven.ndugwa.kabwama@ki.se

## Abstract

COVID-19 vaccines significantly reduced COVID-19 related morbidity and mortality. Although purposeful response measures like movement restrictions affected delivery of other health services, few studies have investigated the association between COVID-19 vaccination as a response strategy and continuity of other immunization services. We aimed to assess the association between the COVID-19 vaccine rollout and continuity of routine immunization services and describe the interventions instituted to maintain delivery in Uganda. This was a cross-sectional study conducted in Wakiso District, Central Uganda. We applied an explanatory, sequential mixed-methods design. We analyzed routine childhood immunization data by computing the percentage change in vaccine doses given for Bacille Calmette-Guerin (BCG), Diptheria, Tetanus Toxoid Pertussis (DPT3), Polio 0, Polio 1, Polio 2 and Polio 3 between March 2021 and April 2021. This was followed by 19 interviews with health workers and 3 focus group discussions with altogether 33 mothers using the World Health Organization Health System Building Blocks as a guiding framework. We found that the COVID-19 vaccine rollout was associated with changes in the trends of routine vaccine uptake. The number of DPT3 vaccine doses reduced by 4.3% between March 2021 and April 2021 after the COVID-19 vaccine rollout while that for Polio 1 vaccine doses reduced by 5.5%, Polio 2 vaccine doses reduced by 5.8% and Polio 3 doses reduced by 5.6%. The challenges to

**Data availability statement:** The aggregate data used for the quantitative analysis as well as the codes and themes that emerged from the qualitative analysis have been provided within the Excel file as Supplementary Material.

**Funding:** Financial support for this study was obtained through a research grant to RKW from the Bill and Melinda Gates Foundation (INV-019313). The views, opinions, and content of this publication are those of the authors and do not necessarily reflect the views, opinions, or policies of the Bill and Melinda Gates Foundation. The funders had no role in study design, data collection and analysis, decision to publish, or preparation of the manuscript.

**Competing interests:** The authors have declared that no competing interests exist.

continuity included increased workload, competition for cold chain and storage capacity and impact on perceptions about vaccination. Interventions to sustain demand included engaging community health workers, community mobilization, health education, and prioritizing routine immunization services. Interventions to maintain delivery included integration of services, increasing health workforce and separating resources for routine vs COVID-19 vaccination. In conclusion, the COVID-19 vaccine rollout was associated with infrastructural and logistical challenges which affected delivery of routine immunization services. Introduction of the COVID-19 vaccines was also associated with negative perceptions about routine childhood vaccines. Deployment of new medical countermeasures should integrate interventions to predict and mitigate effects on existing supply systems like the human resources and infrastructure. Medical countermeasures deployment should also involve education and sensitization that addresses misconceptions and sustains demand for existing health services.

## Introduction

The COVID-19 pandemic was a public health emergency of international concern for more than three years [1] and caused significant morbidity and mortality [2,3]. In response, vaccines were swiftly developed as an important medical countermeasure to reduce COVID-19 related mortality and morbidity. The deployment of medical countermeasures is an important operational function of an emergency response and is critical for minimizing mortality and morbidity during large scale public health events [4]. COVID-19 vaccination outside of study settings started in December 2020 [5,6] and by January 2023, more than 13 billion doses of COVID-19 vaccines had been administered [2] globally. The association between vaccination and reductions in COVID-19 related mortality and morbidity is unequivocal. By March 2023, a retrospective surveillance study showed that COVID-19 vaccination programs in the World Health Organization (WHO) European Region averted 1.6 million deaths, particularly among older age groups [6]. However, these achievements have not been without challenges. National vaccine deployment programs have had challenges such as supply shortages, insufficient funds to facilitate roll out, concerns about vaccine hesitancy and safety as well as disruption of routine essential services [7,8]. Unanticipated impacts of public health interventions have been previously documented [9,10] and have informed the development of the theory of unanticipated consequences of purposive social action [11]. According to the theory, interventions to promote public health can have both foreseen and unforeseen consequences. The theory emphasizes the importance of evaluating public health interventions to identify negative unintended effects so that these can be addressed to maximize the intended benefits of the intervention. Vaccination is indeed an important medical countermeasure to control the spread of COVID-19, and although several studies have shown that purposeful response measures like movement restrictions affect the delivery of other health services [12,13], none, to the best of our knowledge, has investigated the association between COVID-19 vaccination as a response strategy

and the continuity of other immunization services in low-income settings. We conducted a study to assess the association between the COVID-19 vaccination rollout and continuity of routine immunization services and describe the interventions implemented to sustain delivery of routine services. Findings and lessons will inform strategies to minimize disruption to routine services during medical countermeasures deployment in responses to public health emergencies particularly in low- and middle-income countries with resource constrained health systems.

## Methods

### Study area

We conducted the study in Wakiso District in Central Uganda. The district surrounds the capital city Kampala and has the highest district population of over 3 million residents and more than 750,000 households [14]. The households are located across rural, peri-urban and urban contexts which improves the transferability [15,16] of study findings to similar contexts.

### Study design

We conducted the study using an explanatory sequential mixed methods cross sectional design [17]. First, we analysed routinely collected data to assess the magnitude of change in the number of vaccine doses given in Wakiso District between March 2021 and April 2021. We then supplemented this strategy with qualitative interviews with key informants and focus group discussions with service users to elaborate, enhance and clarify [17] the findings from the quantitative results.

### COVID-19 vaccination in Uganda

The COVID-19 vaccine rollout started in March 2021 and prioritised groups such as persons 50 years and older, those with underlying health conditions, health workers, teachers and security personnel [18]. Initially, the vaccines were offered at specific public health facilities. After the first five months of the launch of the vaccination program, the Ministry of Health started accelerated mass vaccination campaigns where COVID-19 vaccines were offered at public places such as places of worship, work places, markets and community outreach.

### Data collection

The sections below provide a detailed description and rationale for the quantitative and qualitative data collection procedures to enhance the dependability [15,16] of study findings.

**Quantitative data.** We obtained routine childhood immunization data, including the number of BCG vaccine doses, DPT3 doses, and Polio doses, from the national health management information system (HMIS) using the District Health Information Software 2 (DHIS2) [19]. Through the DHIS2, health facilities submit weekly and monthly reports about health service use to the national health management information system. The system has been described in detail in previous publications [20–22]. For example, a health facility will compute the total number of BCG vaccine doses given in one month and submit this as an aggregate figure to the national system. The data obtained were for the periods before the COVID-19 pandemic (January 2018 – February 2020), during the COVID-19 pandemic (March 2020 – February 2021) and after the roll-out of the COVID-19 vaccines (March 2021 – December 2022). We used the 2020 [23] and 2021 [14] Annual Statistical Abstracts to estimate the number of children aged 0–4 years in Wakiso District.

**Qualitative data.** The district health data manager guided the purposeful selection of high-volume health facilities at various levels of the health system by comparing monthly outpatient department visits per facility. The first author and a female research assistant with master's degree level training conducted the interviews and focus group discussions. Interviews were conducted with key informants at the health facilities who were involved in the managerial, delivery and logistical operations of immunization services. This included health facility managers, immunization focal persons, nurses

Global Public Health

as well as facility cold chain technicians. Data were collected using an interview guide that was developed using the WHO Health System Building Blocks [24] as a guiding framework. The framework is used to describe a health system by disaggregating it into 6 components, i.e., leadership and governance, service delivery, health system financing, health workforce, medical products, facilities and technologies as well as health information systems. We also included the population to obtain patient perspectives about service delivery as has been recommended by the Health Systems Dynamic Framework [25] – an extended version of the WHO Health System Building Blocks. The interview guide included open questions to elicit information related to the interventions that health workers implemented to promote the uninterrupted delivery of routine immunization services within the health facility. We took field notes during the interviews and focus group discussions to supplement and add nuance to the coding and generation of themes. Interviews were conducted until no new information was being generated. The interview guides and consent forms were translated to Luganda – the locally spoken language in the district.

To improve the credibility or truth value [15,16] of our research, we also conducted 3 key focus group discussions (FGD) to obtain the demand perspectives of the association between the COVID-19 vaccine rollout and perceptions about routine vaccination. We based this estimate on the concept of information power [26] where we considered the homogeneous sample of participants, the number of participants per focus group discussion, the narrow focus of the interview guide and the depth and quality of the discussions to conclude that an additional FGD would not have yielded information that would not already have been captured in the first 3 FGDs. To enhance the trustworthiness of the study in terms of credibility and confirmability [15,16], focus group discussions were conducted after the key informant interviews to provide clarity about the opinions provided by key informants regarding the association between the COVID-19 vaccine rollout and perceptions among patients. The key informant interviews took an average of about 45 minutes while the focus discussions took an average of about 1 hour. The key informant interviews and focus group discussions were conducted in a language comfortable for the participants.

The participants in the focus group discussions were recruited through health workers at three health facilities within Wakiso District. Women were eligible to participate if they had at least 2 children who were less than 3 years old. The rationale for this criterion was to include mothers who received services before, during and after the COVID-19 pandemic.

## Data management and analysis

**Quantitative data.** First, we computed the percentage changes in the number of vaccine doses administered for the various antigens between March 2021 and April 2021. We then used interrupted times series analysis [27,28] using R software to graphically illustrate the interruptions and corresponding changes in the trends and trajectories of service use. Aside from the visual inspection of the trends, the coefficients quantifying the trends in service use from the interrupted times series analysis and poison regression modelling have been provided as S2 File.

**Qualitative data.** A voice recorder was used to record all interviews which were later transcribed verbatim. All analyses were initially done manually following a thematic approach. We employed an abductive approach to data analysis [29] which involved both deductive and inductive processes. This means that deductively, we developed a coding matrix to generate themes following the WHO Health Systems Building Blocks [24] as a guide, but were flexible and open to include themes beyond this framework. To enhance the confirmability of study findings [15,16] the coding matrix was developed prior to the analysis to provide structure to the identification of themes, codes and patterns.

## Ethical considerations

The study protocol was approved by the Higher Degrees Research and Ethics Committee of Makerere University School of Public Health (Protocol #903) and registered with the Uganda National Council for Science and Technology (UNCST) (Approval #HS1121ES). In addition, we obtained permission to conduct the study from the District Health Office of Wakiso District Local Government. Each key informant provided both oral and written informed consent to participate in the study.

## Inclusivity in global research

Additional information regarding the ethical, cultural and scientific considerations specific to inclusivity in global research is included as Supplementary Information (S1 Checklist)

## Results

### Quantitative Results

Across the indicators analysed, the start of the COVID-19 pandemic was associated with an immediate and sharp drop in the number of vaccine doses administered. In addition, the introduction of COVID-19 vaccines was associated with a slower rate of monthly increases in the numbers of vaccine doses administered across all three time periods (Figs 1 and 2).

The COVID-19 vaccine rollout between March 2021 and April 2021 was associated with a 4.3% reduction in the number of DPT3 doses, 5.5% reduction in the number of Polio 1 doses, 5.8% reduction in the number of Polio 2 doses and 5.6% reduction in the number of Polio 3 doses (Table 1).

Beyond the immediate reduction of service use associated with the COVID-19 vaccine rollout, there was a sustained steady recovery of service use across all the indicators.

### Qualitative results

A total of 52 individuals participated in the qualitative part of the study. This included 19 key informants and 33 participants in the focus group discussions. Key informants included immunization focal persons, midwives, nurses as well as cold-chain technicians (S1 File). Of the 19 key informants, 17 (89%) were females and 2 (11%) were from private-for-profit

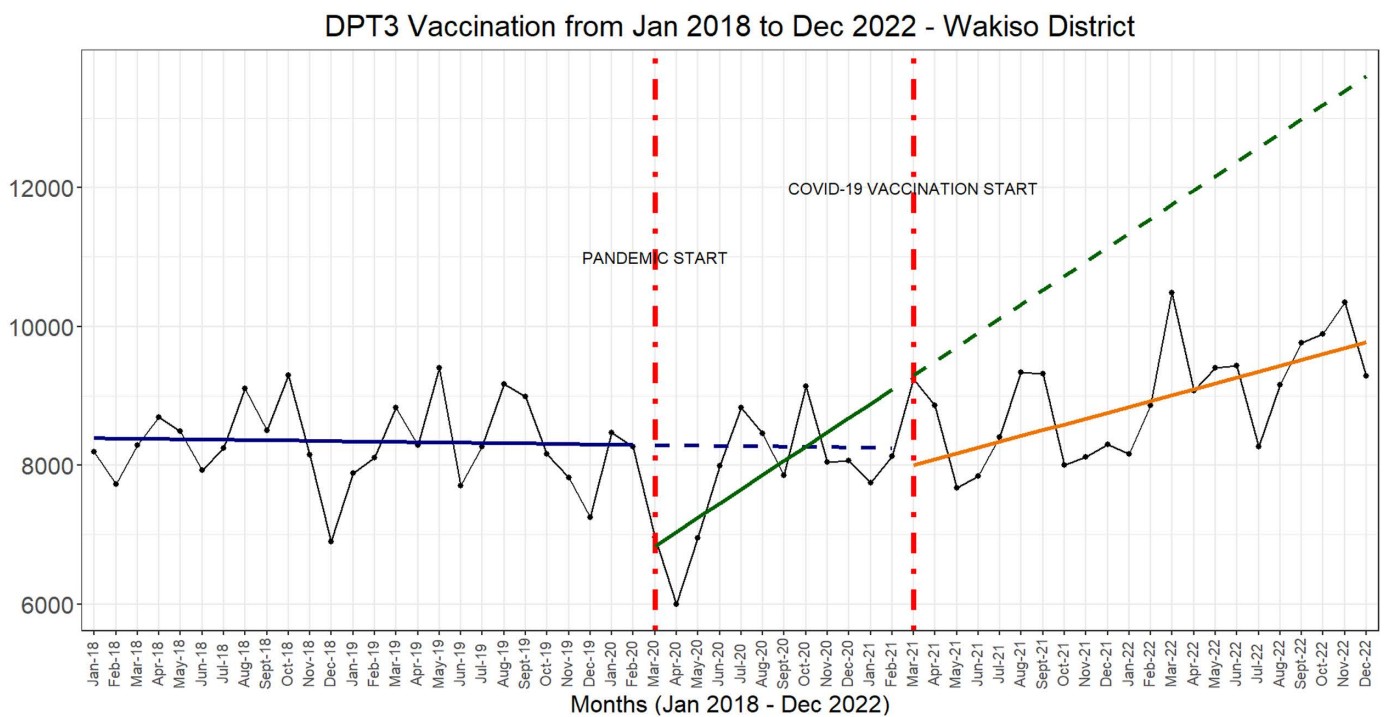

**Fig 1. Interrupted times series graphs for DPT3, BCG and Polio 0 vaccines.**

facilities. The average age for the participants in the focus group discussions was 26 years for FGD one (10 participants), 26 years for FGD two (11 participants) and 26 years for FGD three (12 participants).

The findings from the key informant interviews and focus group discussions covered three broad themes that are summarized in Fig 3. These include findings related to challenges to continuity of routine immunization services, sustaining demand for immunization services and leveraging existing resources to ensure that routine immunization services continued alongside the roll-out of the COVID-19 vaccination.

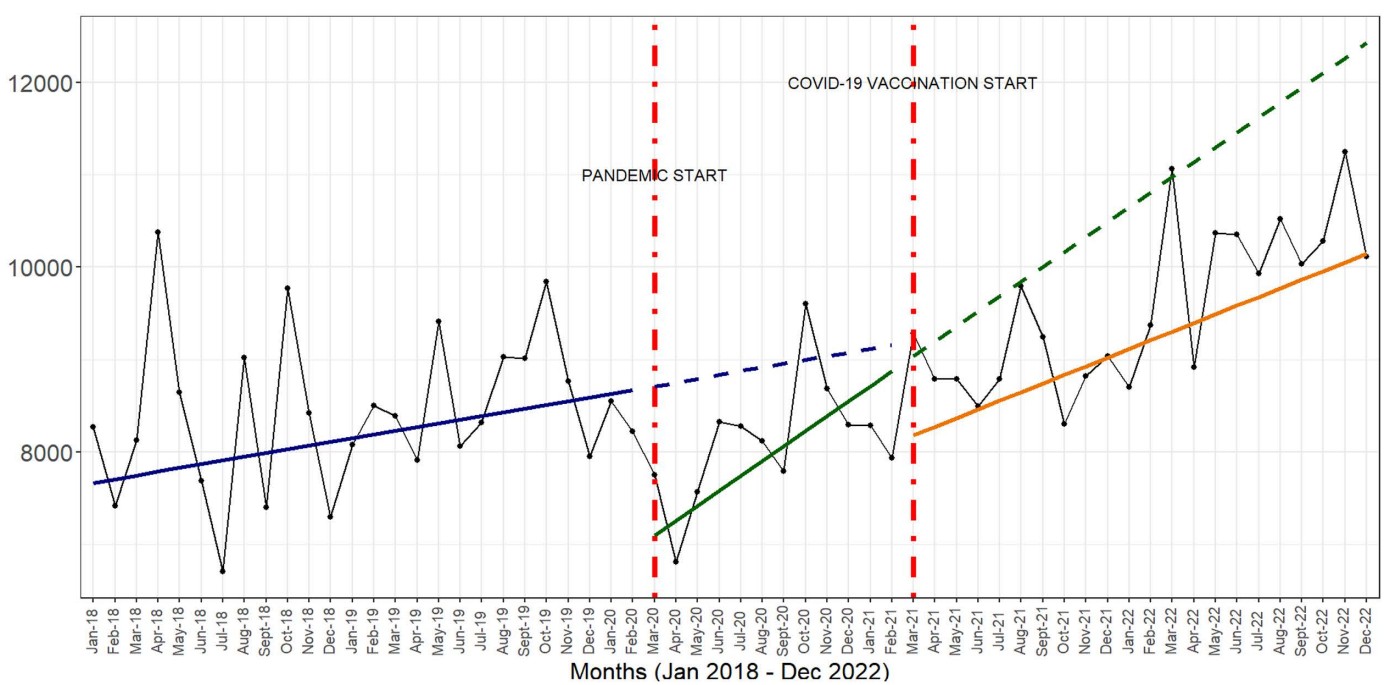

**Fig 2. interrupted times series graphs for Polio 1, Polio 2 and Polio 3 vaccines.**

**Table 1. Percentage changes in service use for various vaccine at the start of the pandemic and the start of COVID-19 vaccination.**

| Vaccine | Pandemic Start | | | COVID-19 Vaccination Start | | |
|---|---|---|---|---|---|---|
| | March 2020 | April 2020 | | March 2021 | April 2021 | |
| Population (0–4 years) | 510160 (%) | 510160 (%) | %Change | 543498 (%) | 543498 (%) | %Change |
| DPT3 | 6966 (1.37) | 5994 (1.17) | -16.2 | 9246 (1.70) | 8862 (1.63) | -4.3 |
| BCG | 6473 (1.27) | 5738 (1.12) | -12.8 | 7235 (1.33) | 7676 (1.41) | 5.7 |
| Polio 0 | 5333 (1.05) | 4890 (0.96) | -9.1 | 6174 (1.14) | 6516 (1.20) | 5.2 |
| Polio 1 | 7752 (1.52) | 6812 (1.34) | -13.8 | 9281 (1.71) | 8793 (1.62) | -5.5 |
| Polio 2 | 7105 (1.39) | 6041 (1.18) | -17.6 | 8973 (1.65) | 8481 (1.56) | -5.8 |
| Polio 3 | 7224 (1.42) | 5865 (1.15) | -23.2 | 9263 (1.70) | 8773 (1.61) | -5.6 |

DPT: Diptheria, Tetanus Toxoid Pertussis, BCG: Bacille Calmette-Guerin

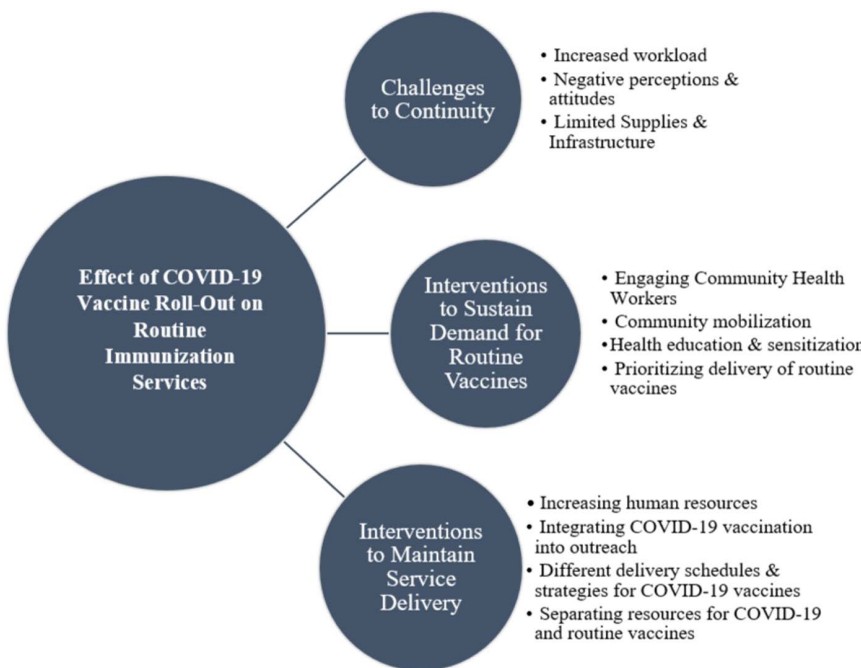

**Fig 3. Broad Themes that Emerged from the interviews.**

### Challenges to continuity of immunization services

**Increased workload.**  Participants noted that the major challenge to continuity of routine immunization services during the COVID-19 vaccine rollout was the increased amount of work required to deliver both sets of services. Health workers noted that the workload became overwhelming as it involved first providing information about the vaccines, obtaining written consent, administering the vaccines and then filling out vaccination cards after the procedure. The human resource challenges were particularly around the need for capturing data from vaccine recipients both at health facilities and during outreach. The fact that a few public health centres provided COVID-19 vaccination services exacerbated the human resource challenges at these specific facilities.

*"…the number of the health workers in the health facility was reduced in the days of the outreach. Now the nurse who had remained in the facility would find themselves having to work on covid immunisation and at the same time routine immunisation. So, you could find that they could be overworked as the responsibilities will have greatly increased."* KII Three

**Limited supplies and infrastructure.**  The COVID-19 vaccine rollout was associated with increased requirements for infrastructure and supplies such as cotton wool, icepacks, cooling boxes, space and vaccine storage capacity. Health workers from several private-not-for-profit facilities noted that additional fridges were procured after assessment of capacity for vaccine storage. Several health workers also noted that the District Local Government provided them with additional fridges for storing vaccines. Some health workers noted that they leveraged available storage capacity and vaccine carriers from nearby health facilities. Other health workers noted repurposing existing storage capacity from the laboratory. In addition, where facilities lacked storage capacity, vaccines were brought directly from the district vaccine stores for use at the health facilities or during outreach. Existing storage capacity was also leveraged through regulating

the quantities of vaccines ordered. For example, health workers switched to requesting routine vaccine quantities for 1 month instead of 3 months to accommodate the additional storage requirements for the COVID-19 vaccines.

*"In terms of storage, time came when our fridge wasn't enough and we had to reduce on the number of vaccines we were requesting for routine. Because with the routine, we would request for vaccines and spend three months without requesting for more. We had to reduce to make sure that you had what could take a month and we had to continue monitoring the stock so that if you see that the stock is going low, that is when you would go and pick some other vaccines."* KII Two

**Negative perceptions and attitudes.** Health workers reported that the COVID-19 vaccine rollout was negatively associated with perceptions about routine immunizations among the public. They noted that although there was an initial reluctance to receive COVID-19 vaccines, this did not negatively impact people's attitudes towards receiving routine vaccines.

*"Okay at first people thought that since we are starting to do covid vaccination, the other routine immunization would stop which wasn't the case, we integrated the COVID-19 vaccination into the routine immunization"* KII Fifteen

This was corroborated by participants in the focus group discussion, albeit with additional insights. Participants in the focus group discussions noted that although COVID-19 vaccination did not negatively impact perceptions and attitudes towards receiving routine immunization, some mothers had developed negative attitudes towards receiving vaccines during mass campaigns and community outreach. Mothers preferred to receive vaccines at health facilities because it was easier to follow up with a health worker in case their child developed an adverse effect after immunization compared with when vaccines are received during mass campaigns and outreach.

*"…parents are afraid of having their children vaccinated during the campaigns… there's hardly anyone accountable for the vaccination that is administered outside the health facility. So, no one can be blamed – in case of adverse effects of the vaccine."* Participant, FGD Two

**Interventions to sustain demand for routine immunization services.** The demand for routine immunization services was sustained through engaging the community health workforce, conducting community mobilization, offering health education and dispelling myths about COVID-19 vaccines and prioritizing routine immunization services at health facilities.
**Engaging community health workers.** The demand for routine immunization services was sustained through partnering with community health workers when planning for community outreach to mobilize mothers with infants to bring them for vaccination services. Community health workers also conducted defaulter tracing which involved looking for any children that had missed their vaccination and then informing them when to obtain the vaccines if outreach had been scheduled or to obtain them from a health facility. Health workers noted that this strategy was effective because of the high level of trust the public has in the community health workers.

*"the [community health workers] help so much in looking for the children who are lost due to follow up. If you have a register and there are children who are scheduled for some antigen and they are not there, sometimes calling them becomes a challenge. But once you involve the VHTs, they know where these children stay and they can easily trace them for you."* KII Seventeen

**Community mobilization.** Health workers conducted community mobilization using public address systems to encourage the public to continue coming to health facilities for immunization services. The mobilization was done by

community leaders such as local council leaders, church leaders and community health workers. Community health workers who had megaphones also used these to promote continuity of uptake of routine immunization services. Local leaders and community health workers also used local radios to encourage the public to continue seeking routine immunization services.

*"we would… pay a public address system to voice information in the community telling them that we are going to do this and not COVID… we would give the information to the (village leaders) and they pass it through the churches and then the churches would pass it on to the community members. And then also the use of village health teams within the community. So, they are the ones we could first tell that you we are not carrying out covid vaccination today but it's basically for babies or these normal outreach where we give vitamins to the young ones, deworming tablets to the babies."* KII Eighteen

**Health education and sensitization.**  Health education was also an important strategy for sustaining demand for routine immunization services. Before immunization, health workers gave education sessions covering issues such as the vaccine preventable diseases, injection sites, possible side effects, the benefits and importance of completing the vaccination schedule and the phone numbers to call in case of any adverse effects after immunization. Health workers also used this opportunity to provide mothers with information about COVID-19 vaccination including benefits of the vaccine, possible side effects and responding to questions and dispelling myths and falsehoods about COVID-19 vaccination. This strategy was effective in sustaining the demand for routine immunization services and promoting the uptake of COVID-19 vaccines.

*"I will talk about this vaccine. I can decide to talk about the rota virus. I will tell the parents what it means, the benefits, and if the child doesn't get this, what are the implications… we even encourage them to tell their neighbours to bring their children for immunisation… we even talk of the side-effects"* KII Two

**Prioritizing routine immunization at health facilities.**  The demand for immunization services was also sustained through prioritizing the delivery of routine immunization services over COVID-19 vaccination services. In circumstances where both services were delivered on the same day, children received routine immunization services first and then adults were offered COVID-19 vaccination services after. Other health facilities designated the first half of the day to routine immunization services and the afternoons for COVID-19 vaccination services. In other cases, health workers increased the number of days when routine immunization services were offered.

*"we used to allocate to them days when we were sure that we weren't going to do routine immunization. Even when a person came and wanted to immunize against COVID, you would tell them that today is not for COVID, we are vaccinating the children."* KII One

**Interventions to maintain delivery of routine immunization services.**  The routine delivery of immunization services was maintained through leveraging existing infrastructure and supplies, modification of service delivery and sourcing for additional human and financial resources.

**Different delivery schedules and strategies for COVID-19 vaccination.**  To ensure uninterrupted delivery of routine immunization services, health workers adopted different service delivery strategies for COVID-19 immunization services. These included the use of mobile clinics, conducting COVID-19 specific outreach in places like markets, mosques and churches and using door to door strategy. The rationale for adopting different delivery strategies was to prevent crowding in health facilities since the target population for COVID-19 vaccines were adults who could receive these services outside of a health facility setting.

*"we carried out covid-19 campaigns; for example, we went to (entertainment venues), we went to the markets, the churches, and the communities…"* KII eighteen

**Integration of COVID-19 vaccination into routine outreach.** Health workers leveraged existing resources by integrating the COVID-19 vaccination into the outreach for Polio, Measles and other vaccines routinely conducted within communities. This further enabled the uninterrupted delivery of both the static services within health facilities but also the routine outreach, albeit with additional services for persons who wanted to receive COVID-19 vaccines.

*"We decided to incorporate COVID vaccination into the routine outreach. When we would go for routine outreach, we would take the COVID vaccines and vaccinate the women from their villages."* KII One

**Increasing health workforce.** Health workers noted that the rollout of the COVID-19 vaccination came with increased demands on the health workforce to support both the routine delivery of immunization services and the COVID-19 vaccination. This was addressed through temporary recruitment of student nurses, interns and volunteers to support the additional data entry requirements. Health implementing partners like Red Cross also provided volunteers to support the COVID-19 vaccine rollout. In other cases, health workers from private clinics were trained and recruited to support activities such as recording and vaccination during outreach for both COVID-19 and routine immunization.

*"For covid vaccination, there were some health workers who had the clinics, we would work with them. When we had an outreach in a certain area, we would work with a health worker who has a clinic in that area. There was a lot of paper work so, the health workers in the clinics around the areas would help us to make sure of both vaccinations carry on smoothly."* KII Thirteen

**Separating COVID-19 vaccination resources from routine immunization resources.** The Ministry of Health provided separate financial and material resources to prevent reallocation of resources away from routine immunization services. For example, supplies such as cooling boxes, diluents, syringes, safety boxes and patient registers were separately provided to facilitate COVID-19 vaccination. The Ministry of Health also provided allowances to health workers providing COVID-19 vaccination services especially during outreach. In addition, implementing partners such as Save the Children provided financial support to facilitate movement of health workers during outreach for COVID-19 vaccinations.

*"…they are different. Covid vaccination funds come differently and so are the funds meant for routine immunisation. The money for routine immunisation comes from PHC (Primary Health Care) but that of covid vaccination comes from a different source."* KII Seven

## Discussion

This study showed that the COVID-19 pandemic and the subsequent COVID-19 vaccine rollout were associated with declines in trends and trajectories of routine immunization services at health facilities. Our results are consistent with previous global [30] and national assessments in Uganda [31] and other low and middle-income countries [32,33] that found that the COVID-19 pandemic was associated with challenges such as fear of infection in health settings, reluctance to access services, reduced household income and movement restrictions which negatively affected access to and delivery of other routine health services. These challenges could explain the immediate reduction in service use associated with the COVID-19 vaccine rollout observed across the indicators we analysed. The COVID-19 vaccine rollout was also associated with increased workload, reduced storage capacity and negative perceptions about routine childhood vaccines. The finding of increased workload supports earlier projections that estimated that a 112.7% increase in workload would be

required to deliver COVID-19 vaccines to high risk groups [34]. In our study, the increased demand on the health workforce was addressed through ad hoc measures such as recruiting student nurses and interns, volunteers and "borrowing" health workers from private health providers. Immunization programs should be deliberate about increasing vaccinators through strategies such as establishing formal agreements with private healthcare providers to support immunization service delivery for both COVID-19 vaccines and routine vaccines. Such agreements and frameworks are critical for optimizing the contribution of the private sector to emergency response while achieving both public and private sector interests [35,36].

Our study also found that the COVID-19 vaccine rollout had a peculiar association with perceptions and attitudes towards routine vaccination – particularly regarding where vaccines are received. Mothers noted that generally, they were not any less likely to vaccinate children due to perceptions they had about COVID-19 vaccines but were less likely to vaccinate children during outreach and campaigns. The literature on the impact of COVID-19 vaccination on perceptions about routine immunizations is mixed. Some studies found that the COVID-19 vaccine rollout was correlated with increased vaccine hesitancy [37,38], others found no association with perceptions [39] while others found improved confidence in routine vaccines after the COVID-19 vaccine rollout [40]. Even where vaccine hesitancy increased, it did not translate to decreased intention to vaccinate with routine childhood vaccines [37]. The mixed findings may be attributed to contextual differences in perceptions about vaccine effectiveness, parental perceptions about health, beliefs about childhood diseases and levels of trust in public health institutions that conduct the vaccination [41]. In our study, mothers noted that since the COVID-19 vaccine rollout, they preferred to vaccinate their children at health facilities instead of community outreach. This finding could be attributed to increases in anxiety about the side effects of COVID-19 vaccines as has been noted elsewhere [39]. This finding also corroborates results from the Uganda Health Sector Performance report [42] that highlighted the persistence of myths, misconceptions and misunderstandings about vaccines and the need to address these issues to improve uptake in the general population. Indeed, one of the strategies employed for sustaining demand for immunization services at health facilities was health education and sensitization conducted by health workers to mothers before vaccinating their children. The education sessions covered issues such as the vaccine preventable diseases, injection sites, possible side effects, the importance of completing the vaccination schedule among others. For vaccination outside of health facility settings, scholars have recommended that social mobilization is critical to increase vaccine confidence [43,44]. It involves paying attention to the message (importance of vaccines, addressing misconceptions), the messenger (health workers, trusted community members) as well as engaging communities when designing messages to improve uptake. Health workers noted that they sustained demand through conducting health education at facilities and community mobilization. These strategies should involve communities to develop effective social mobilization campaigns ahead of mass campaigns and outreach.

The association between the COVID-19 vaccine rollout and reductions in existing storage capacity is not a surprise finding. Prior to the pandemic, many low- and middle-income countries had inadequate and unreliable cold chain and transport capacity [45]. Conservative estimates of additional storage requirements for COVID-19 vaccines were estimated to be about 5% at national level and 2% at subnational level [34]. In our study, storage capacity was expanded through procuring additional fridges, repurposing existing storage facilities, using vaccine stores at higher levels of the health system and regulating quantities of routine vaccines ordered. These innovations could explain the recovery of services seen in the interrupted times series analysis across the indicators analysed. In addition to formalizing strategies to increase the health workforce for campaigns, immunization programs should develop guidance for improving and expanding cold chain capacity during medical countermeasures deployment in emergency response. For example, clear policy and guidance were provided to successfully outsource vaccine cold chain and transport management to the private sector in South Africa [46].

Sustaining demand through engaging community health workers [47–49] and maintaining service availability through integrating health services [49–51] have both been highlighted by other researchers as important strategies for service

continuity during the COVID-19 pandemic. Other study findings include the importance of prioritizing routine immunization services and separating resources (diluents, syringes, cooling boxes e.t.c) meant for COVID-19 vaccination from routine vaccination. This requires vaccination management systems that minimize competition for time, space, and other resources between the two. This should be a coordinated effort between policy makers at national level and implementers at regional and health facility levels.

Findings from this study should be interpreted in light of a few limitations. First, the quantitative analysis was based on routinely collected data – the completeness and quality of which are sometimes poor. However, the Ministry of Health conducts regular data quality audits to improve the quality of these data. The other limitation is related to the interpretation of the quantitative analysis. The computation of the percentage changes in service use at the start of the pandemic and the start of the COVID-19 vaccine rollout limits the interruption to a specific period and yet these disruptions could occur over extended periods. However, we have provided supplementary material that includes the results of the interrupted times series analysis that shows trends in service use before, during and after the COVID-19 vaccine rollout.

## Conclusions

The COVID-19 vaccine rollout was associated with changes in trends and trajectories of routine vaccine uptake. In addition, the rollout affected human resource and vaccine delivery infrastructure in health facilities and affected people's perceptions about routine vaccines. The rollout of medical countermeasures during public health emergencies should acknowledge the demand that such response strategies have on workload and infrastructure, and develop policy guidance that enables programs to leverage support from the private sector. Risk communication messaging during emergency response should also integrate social mobilization to improve the public's awareness and appreciation of new medical countermeasures. There is also a need to develop coordination and management systems that minimize competition for time, space, human, financial and other resources between new medical countermeasures and routine services.

## Supporting information

**S1 File. Details of key informants.**
(DOCX)

**S2 File. Interrupted times series and poisson regression analysis outputs.**
(DOCX)

**S1 Checklist. Inclusivity in global research questionnaire.**
(DOCX)

**S1 Data. Vaccination code book.**
(XLSX)

## Acknowledgments

We thank all individuals who participated in this study. We are also grateful to Amina Nambuusi who diligently supported the qualitative data collection process.

## Author contributions

**Conceptualization:** Steven Ndugwa Kabwama, Neda Razaz, John M Ssenkusu, Helena Lindgren, Rhoda K. Wanyenze, Tobias Alfvén.

**Data curation:** Steven Ndugwa Kabwama.

**Formal analysis:** Steven Ndugwa Kabwama.

**Funding acquisition:** Rhoda K. Wanyenze.

**Investigation:** Helena Lindgren, Rhoda K. Wanyenze, Alfred Driwale, Tobias Alfvén.

**Methodology:** Steven Ndugwa Kabwama, Neda Razaz, John M Ssenkusu, Helena Lindgren, Tobias Alfvén.

**Project administration:** Steven Ndugwa Kabwama, Rhoda K. Wanyenze.

**Resources:** Rhoda K. Wanyenze.

**Supervision:** Neda Razaz, John M Ssenkusu, Helena Lindgren, Rhoda K. Wanyenze, Tobias Alfvén.

**Validation:** Neda Razaz, John M Ssenkusu, Helena Lindgren, Alfred Driwale, Tobias Alfvén.

**Writing – original draft:** Steven Ndugwa Kabwama.

**Writing – review & editing:** Neda Razaz, John M Ssenkusu, Helena Lindgren, Rhoda K. Wanyenze, Alfred Driwale, Tobias Alfvén.

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
