## [Decision Letter · Decision Letter 0]

PGPH-D-25-00240

Maintenance of Service Delivery During Medical Countermeasures Deployment: The Effect of the COVID-19 Vaccine Roll Out on Routine Childhood Immunization Services in Uganda

Dear Dr. Kabwama,

Thank you for submitting your manuscript to PLOS Global Public Health. After careful consideration, we feel that it has merit but does not fully meet PLOS Global Public Health’s publication criteria as it currently stands. Therefore, we invite you to submit a revised version of the manuscript that addresses the points raised during the review process.

EDITOR: 

The authors should ensure that the methodological limitations of the study are fully addressed, preferably in a dedicated paragraph as part of the Discussion section. This should include consideration for any biases introduced and how these were managed. Any impact on interpretation / generalizability of findings should be underscored.

We look forward to receiving your revised manuscript.

Kind regards,

Edina Amponsah-Dacosta, Ph.D., MPH

Academic Editor

Journal Requirements:

Additional Editor Comments (if provided):

Reviewers' comments:

Reviewer's Responses to Questions

**Comments to the Author**

1. Does this manuscript meet PLOS Global Public Health’s publication criteria?

Reviewer #1: Yes

Reviewer #2: Yes

2. Has the statistical analysis been performed appropriately and rigorously?

Reviewer #1: Yes

Reviewer #2: Yes

3. Have the authors made all data underlying the findings in their manuscript fully available (please refer to the Data Availability Statement at the start of the manuscript PDF file)?

Reviewer #1: Yes

Reviewer #2: Yes

4. Is the manuscript presented in an intelligible fashion and written in standard English?

Reviewer #1: Yes

Reviewer #2: No

Reviewer #1: The paper addresses a critical, yet often overlooked, aspect of pandemic response — how the introduction of emergency countermeasures like COVID-19 vaccines can inadvertently disrupt essential routine health services. This study fills an important gap in the literature and is especially relevant for health systems in low- and middle-income countries navigating limited infrastructure, human resources, and competing health priorities.

What Works Well: The use of a sequential explanatory mixed-methods design is appropriate and well-justified. The quantitative data establish clear trends, while the qualitative findings give much-needed depth and context, capturing the lived experiences of health workers and caregivers during the pandemic. I also commend the authors for grounding their analysis in established frameworks (WHO Health System Building Blocks, Health Systems Dynamics Framework), which adds structure and relevance to the findings.

The manuscript is well-written, coherent, and the policy implications are clearly drawn. The topic is timely, the methods are solid, and the findings have both practical and scholarly value.

Suggestions for Improvement:

Although the study is strong, a few revisions could further improve its clarity, precision, and impact.

1. Clarify Language Around Causality

The quantitative analysis — particularly the interrupted time series — is appropriate for exploring associations over time. However, the manuscript occasionally leans into causal language (“impact,” “effect”) which may overstate the findings. I recommend using more cautious terms like “associated with” or “linked to” unless additional modeling or control for confounders is introduced.

2. Reconsider Emphasis on One-Month Comparison

The use of percentage changes between March and April 2021 to demonstrate changes in vaccine uptake is informative but limited. One month is a narrow window, and seasonal or short-term factors could influence results. The ITS graphs provide a more complete picture and should be emphasized more strongly in both the results and discussion. If possible, consider expanding the quantitative interpretation to reflect longer-term trends.

3. Improve Integration Between Quantitative and Qualitative Results

Currently, the quantitative and qualitative findings are somewhat compartmentalized. The discussion would benefit from more explicit connections between the two — for example, linking the observed declines in immunization to specific barriers like cold chain competition, workload pressures, or community hesitancy described by participants.

4. Visual Clarity of Figures

The ITS graphs are helpful but could be improved. Enhancing the resolution, labeling axes more clearly, and adding vertical reference lines (e.g., to mark the vaccine rollout or pandemic start) would make the data more reader-friendly.

5. Data Sharing and Transparency

The Data Availability Statement suggests that data are included as supplementary material. However, it would be good to confirm that the raw data used in the ITS analyses (i.e., monthly vaccine counts) are available in a downloadable format (e.g., Excel or CSV). If sharing full qualitative transcripts isn’t possible for ethical reasons, a brief justification would be helpful.

Minor Notes for improvement:

1. Consistency in Terminology and Style: The manuscript alternates between phrases like “COVID-19 vaccine rollout,” “COVID-19 vaccination,” and “deployment of COVID-19 vaccines.” While all of these are technically correct, consistent use of a single term (e.g., “COVID-19 vaccine rollout”) would improve readability and flow.

Suggested action: Decide on one preferred phrase and apply it consistently throughout the text, including the abstract, results, and discussion sections.

Example:

Line 29: “…the COVID-19 vaccination rollout…”

Line 388: “…the subsequent rollout of COVID-19 vaccines…”

These could both be rephrased to maintain consistency (e.g., use "COVID-19 vaccine rollout" throughout).

2. Reference Formatting and URL Display: Several references (particularly government or WHO documents) contain long URLs that are broken across multiple lines or presented in a cluttered format, which can disrupt the reading experience and reduce professionalism.

Suggested action: Use hyperlink text (e.g., "Uganda Statistical Abstract 2021") instead of pasting long URLs directly. Also, ensure uniform citation formatting (e.g., consistent punctuation, date format, journal names).

Example:

Ref 14: “https://www.ubos.org/wp-content/uploads/publications/01_20222021_Statistical_Abstract.pdf” → could be replaced with: Uganda Bureau of Statistics. (2021). Statistical Abstract. https://www.ubos.org/wp-content/uploads/publications/01_20222021_Statistical_Abstract.pdf

Also ensure references to journal articles include full volume, issue, and page numbers consistently across the list.

3. COREQ Checklist – Transparency on “Not Applicable” Items: The authors have done well to complete the COREQ checklist. However, several items are marked "Not Applicable" (e.g., transcript return, participant checking), and while this may be appropriate, it's important to briefly justify these omissions in the manuscript itself — ideally in the methods section or a brief note in the limitations.

This is especially relevant for qualitative research, where transparency and reflexivity are central to trustworthiness.

Suggestions on How to Address This:

For transcript return:

“Transcripts were not returned to participants for comment or correction due to logistical constraints and time limitations. However, efforts were made to ensure accuracy through verbatim transcription and note-taking during interviews.”

For participant checking of findings:

“Findings were not formally validated with participants after analysis; however, triangulation across multiple data sources and use of a structured coding framework supported the credibility of results.”

These clarifications would reassure readers and reviewers that the researchers were mindful of qualitative rigor, even if certain validation steps were not feasible.

Although these are relatively small issues, addressing them would improve the readability, professionalism, and transparency of the manuscript. They do not detract from the strength of the study but will contribute to its polish and clarity for publication.

Reviewer #2: Interesting study and an important contribution to the literature on how COVID-19 impacted routine immunisation.

Overall, I think the language needs to be made a bit clearer. i.e. outreaches should just be outreach and there are some sentences that are difficult to understand.

Line 110 - underlying 'health' conditions?

Line 122-123 - this doesn't add anything to know that they have been described in pervious papers. I would just stick with the example you have listed and delete this sentence.

In your methods - you need to include whether the facilities were government owned, not-for-profit etc. as you discuss this in your results but I couldn't link it back to your methods.

WHO building blocks - you say you used this to guide the surveys and interviews but then in your analysis you have not used it to frame your results. Where do all the challenges and strategies you have identified fit into the WHO building blocks framework?

**Do you want your identity to be public for this peer review?** For information about this choice, including consent withdrawal, please see our Privacy Policy

Reviewer #1: **Yes: ** Darlington David Faijue

Reviewer #2: No

---

## [Decision Letter · Decision Letter 1]

Maintenance of Service Delivery During Medical Countermeasures Deployment: The Association between the COVID-19 Vaccine Rollout and Continuity of Routine Childhood Immunization Services in Uganda

PGPH-D-25-00240R1

Dear Mr. Kabwama,

We are pleased to inform you that your manuscript 'Maintenance of Service Delivery During Medical Countermeasures Deployment: The Association between the COVID-19 Vaccine Rollout and Continuity of Routine Childhood Immunization Services in Uganda' has been provisionally accepted for publication in PLOS Global Public Health.

Best regards,

Edina Amponsah-Dacosta, Ph.D., MPH

Academic Editor

Reviewer Comments (if any, and for reference):

Reviewer's Responses to Questions

**Comments to the Author**

Reviewer #1: All comments have been addressed

Reviewer #2: All comments have been addressed

publication criteria?

Reviewer #1: Yes

Reviewer #2: Yes

3. Has the statistical analysis been performed appropriately and rigorously?

Reviewer #1: Yes

Reviewer #2: Yes

4. Have the authors made all data underlying the findings in their manuscript fully available (please refer to the Data Availability Statement at the start of the manuscript PDF file)?

Reviewer #1: Yes

Reviewer #2: Yes

5. Is the manuscript presented in an intelligible fashion and written in standard English?

Reviewer #1: Yes

Reviewer #2: Yes

Reviewer #1: Dear Authors,

Thank you for your thorough revision and thoughtful responses to previous comments. The manuscript has improved in clarity, structure, and overall readability. The methods and findings are well presented, and the use of the WHO Health Systems framework continues to provide a strong organizational structure.

I have no additional major concerns. The language is now appropriate for publication, and the paper contributes meaningfully to understanding how public health systems can adapt to medical countermeasures deployment, particularly in low-resource settings.

No further revisions are required from my side.

Reviewer #2: (No Response)

**Do you want your identity to be public for this peer review?** For information about this choice, including consent withdrawal, please see our Privacy Policy

Reviewer #1: **Yes: ** Darlington David Faijue

Reviewer #2: No
